# A time-varying analysis between economic uncertainty and tourism development in Singapore

**Bui Hoang Ngoc**[1]*, **Canh Chi Hoang**[2], **Nguyen Huynh Mai Tram**[3]

**1** Faculty of Business Administration, Ho Chi Minh City University of Industry and Trade, Ho Chi Minh City, Vietnam, **2** Faculty of Business Administration, Ho Chi Minh University of Banking, Ho Chi Minh City, Vietnam, **3** The Post Graduate School, Ho Chi Minh City Open University, Ho Chi Minh City, Vietnam

\* ngocbh@huit.edu.vn

**Data Availability Statement:** All relevant data are within the paper.

**Funding:** The author(s) received no specific funding for this work.

## Abstract

Tourism development (TO) is seen as a viable solution to address economic policy uncertainty (EPU) risks. However, previous studies have largely ignored the relationship between short, medium, and long term by decomposing TO and EPU index at different time-frequency scales, especially in Singapore. In this study, the Wavelet tools analysis and a rolling window algorithm are employed to re-visit the causal relationship between EPU, industrial production index (IPI), government revenue (GR), and tourism development (TO) in Singapore from January 2003 to February 2022. The findings revealed the heterogeneous effects of EPU on TO at different time horizons in terms of importance and magnitude over time. A rise in EPU results in a decline in TO at the low frequencies, indicating that EPU has a detrimental effect on TO over the short term. Conversely, in the long term, an increase in TO results in a decrease in EPU. Furthermore, the outcome also indicated that there is a unidirectional causality running from TO to EPU, GR and IPI. Expressly, we confirm that the negative co-movement is more pronounced in the aftermath of the COVID-19 crisis, particularly for EPU, and GR at low-medium frequencies throughout the research period. The findings provide tourism policymakers with insight to develop strategic plans for tourism development that consider the effects of economic policy uncertainty. By understanding how uncertainty impacts tourism, governments can tailor development strategies to mitigate risks and capitalize on opportunities.

## 1. Introduction

The World Tourism Organization has affirmed that tourism is a major pillar of both developed and developing economies. After fuels and chemicals, tourism has emerged as the third-largest export sector in the world, ahead of food and automobiles [1]. When it comes to the benefits of tourism, it is undeniable that tourism boosts the revenue of the economy, creates thousands of jobs, develops the infrastructure of a country, and plants a sense of cultural exchange between foreigners and citizens [2, 3]. Investments and financial resources are generated from the tourism business, contributing to supporting the activities of local and central state

**Competing interests:** The authors have declared that no competing interests exist.

administrative agencies and, at the same time, improving other issues in society, such as raising living standards. . .

However, the tourism industry is vulnerable to fluctuations such as crises, epidemics, etc. Using the World Uncertainty Index developed by Karabulut, Bilgin et al. [4] had proven that the pandemic has had a huge impact on tourism in low-income countries as well. And it received concurrence from Yang, Zhang and Chen [5], who showed the serious impact of the COVID-19 pandemic on the tourism industry in developed countries such as the US, Spain, and Italy. Therefore, we see that the strong development of tourism as a pillar contributing to economic growth will tend to have the opposite effect, creating economic instability when negative external factors occur. In fact, tourism development and economic instability have a two-way effect and should be well understood. In previous studies, much of the literature looked at factors such as economic uncertainty, political instability, and epidemics affecting the tourism industry in different regions of the world. However, since the outbreak of the COVID-19 epidemic, tourism development has also created economic instability and has not received the attention of academia and government. The question is how tourism development in the countries will cause economic instability.

Based on the context of a strongly developed tourism industry and planned as an important sector generating government revenue, Singapore is considered an attractive example for examining the relationship between tourism and economic instability before and after the COVID-19 pandemic. In 2015, Singapore invested 19.8 billion SGD in the tourism and travel industry, accounting for 19.9% of the total investment in the Singapore economy [6]. In 2020, it contributes 52.5 billion SGD ($40.4 billion), accounting for 11.1% of GDP, and has created around 527,500 jobs, contributing to 14.1% of the country's total workforce [7]. According to figures from the Singapore Tourism Board, international visitor arrivals showed consistent growth with a Compounded Annual Growth Rate (CAGR) of 5.23% for three consecutive periods, reaching a record of 19.1 million in 2019 before declining in 2020 due to the COVID-19 pandemic. Total visitor arrivals decreased by 88.0% from 2020 to 329,970 total visitor arrivals in 2021. Pre-covid, the number of international visitors between 2016 and 2019 showed consistent growth compared to a CAGR of 6.23% between 2008 and 2016. The visitor days decreased from 11.76 million to 7.4 million in 2021 [8].

The above figures show that the Singapore government will face challenges and economic uncertainties rooted in the lockdown and restrictions on movement in and out of the country. Therefore, the complex and multifaceted relationship between economic instability and tourism development has received much attention from economists and policymakers in Singapore. From an economic perspective, tourism's serious influence on economic instability can be seen from three angles: 1) influence on economic policy uncertainty; 2) impact on related industries, and 3) reduce total government revenue. Therefore, the motivations for implementing this study are summarized as follows:

(i) Firstly, the nexus between tourism development and economic policy uncertainty (EPU) has been studied by many researchers, but there is no consistent result. EPU, developed by Baker, Bloom and Davis [9], is used as a proxy for movements in policy-related economic uncertainty. Uncertain economic policies lead to risks to price stability and exchange rate, which affect tourism demand. Specifically, EPU restricts domestic tourism in New Zealand [10], the United Kingdom [11], and Tunisia [12]. In contrast, Nguyen, Thanh and Nguyen [13] concluded that increased economic instability increases domestic tourism and decreases outbound tourism in 124 countries. However, Payne, Topcu et al. [14] found no Granger causality from the EPU to the US outbound travel. Therefore, it also is necessary to investigate this interaction in the context of Singapore.

(ii) Secondly, the labor-intensive nature of the tourism industry creates numerous job opportunities for locals with minimal formal training. Governments generate revenue from tourism through direct and indirect contributions. Direct contributions are taxes levied on tourism-related employment, businesses, and charges on tourists, while indirect contributions result from taxes and duties on goods and services supplied to tourists. Governments at all levels raise tax revenues from the various activities of tourists. The collected revenue finances the goods and services provided to people and businesses and fulfills the government's redistributive role. A comparison of government income levels across countries highlights the importance of the government sector in the economy to allocate resources for developing the tourism sector. This relationship has also been ignored in Singapore.

(iii) Thirdly, tourism is closely related to other industries and significantly impacts the industrial producer index (IPI). Hailemariam and Ivanovski [15] showed that a positive shock in world industrial production leads to an increase in tourism net exports, while a negative shock in industrial production can cause an increase in tourist arrivals. Many industries, including manufacturing, telecommunications, banking, medical and dental care, security, transportation, and lodging, cover tourists' demands. For instance, policymakers recognize healthcare facilities as an essential destination attribute for tourists, and destination marketers must reconsider why healthcare travelers choose to visit a particular location. Alternatively, the demand for hospitality services frequently fluctuates due to seasonal tourist flows. Therefore, the IPI is a critical determinant of a country's economic development and plays a significant role in tourism development, including Singapore.

This study contributes to the literature on three aspects compared to previous studies. *First*, the study fills in the gap in the co-movement, and lead-lag structure relationship between tourism development and economic instability, considering the influence of economic policy uncertainty, related industrial production, and total government revenue on tourism. *Second*, the study is a crucial case study in sustainable tourism development in the era of economic uncertainty in Singapore, a country with a spearhead orientation for tourism development. How will economic instability affect the tourism development of a small island city-state that has become one of the world's most popular tourist destinations? *Finally*, a larger number of available studies employed the time-invariant framework to explore the impact of EPU on tourism development, such as autoregressive distributed lag (ARDL) and vector autoregression (VAR) approaches. However, the restriction of these methods does not capture features across a wide range of frequencies and events that are local in time, which raises doubts about the effectiveness of suggested policy implications [16–18]. To overcome these limitations, in this study, the authors take into account the ability to identify the frequency bands at which significant coherence occurs and the time-varying characteristics of these time series through Wavelet coherence and time-varying causality methods. The advantages of these methods will be presented in-depth in the methodology section.

The rest of the paper is presented as follows: Section 2 is a Literature review, Section 3 outlines the methodology and data used for analysis, empirical results, and discussion are written in Section 4, and the last section is the conclusion and policy implications.

## 2. Literature review

### Economic instability

In the early 1930s, the term "economic instability" was of interest to many economists after crises as well as cyclical downward movements. The term "instability" can be used to describe any repetitive, involuntary, or unpredictable change, positive or negative, in an individual's life,

within a household, or throughout the economy [19]. The consideration of instability is often based on different aspects such as economic, cultural, and political, and is closely watched over a short period through the frequency, predictability, and duration of changes (e.g. yearly, weekly, monthly). Researchers have studied economic volatility by measuring changes such as coefficient of variation or percent of income change and to what extent or type of change constitutes a significant change [20–23]. Similarly, Hill, Romich et al. [22] assumed that economic instability is repeated changes in employment, income, or financial status over time, especially unintended, predictable changes. Economic instability can have several negative effects on the general welfare of people and countries by creating an environment in which economic assets lose value and investment is impeded or stopped. This can lead to unemployment, economic recession, or in extreme cases, social collapse.

Western, Braga et al. [24] discussed how often economic instability stems from adverse events such as job loss, family breakdown, and poor health. In particular, it is also partly derived from the macroeconomic policies that have been implemented in each country as these policies are distorted, which can cause higher inflation rates, skewed real exchange rates, and unsustainable fiscal deficits. In this article, economic instability is defined as the negative and unforeseen turmoil of a country's economic situation and measured with three factors: uncertainty of economic policy, industrial production index, and total revenue of government activities.

Previous literature has rarely focused on observing and understanding the impact of tourism development on economic instability. This research gap should be discussed because when COVID-19 broke out, travel became difficult to consider as one of the causes that seriously affected all aspects of the economy. In particular, for countries that focus on strongly developing tourism and travel industries, the decline in tourism revenue continuously impacts related industries such as retail, restaurant-hotel. . . and government revenue. Therefore, studying the impact between tourism development and economic instability is necessary.

In this study, the authors use economic policy uncertainty as a proxy for economic instability. Economic policy uncertainty refers to the uncertainty surrounding government policies and regulations related to the economy, including fiscal, monetary, trade, and regulatory policies. Changes or uncertainty in these policies can directly impact economic behavior, including investment decisions, consumption patterns, and business planning. High levels of policy uncertainty can lead to hesitancy among businesses and consumers, which can contribute to economic instability. Besides, research has shown that economic policy uncertainty is often correlated with other measures of economic instability, such as stock market volatility, business investment, and GDP growth rates. High levels of policy uncertainty tend to coincide with periods of economic downturns, recessions, or financial crises, suggesting that it can serve as a useful proxy for broader economic instability.

## Tourism development and economic policy uncertainty

The relationship between tourism development and economic policy uncertainty is explained through the Tourism risk perception theory developed by Yang and Nair [25]. This theory focuses on how individuals perceive and respond to risks associated with travel and tourism activities [26]. Perceiving risks in tourism entails three perspectives: subjective emotions, objective assessment, and understanding the potential for negative outcomes beyond a certain threshold during travel. Subjective aspects of risk perception involve personal and psychological elements, while objective factors encompass physical, economic, social, psychological, temporal, and opportunity risks. Economic policy uncertainty can amplify perceived risks associated with travel, such as job loss, financial strain, or health concerns [27]. Economic

policy uncertainty can impact consumer confidence and spending patterns, which are crucial drivers of tourism demand. Uncertainty about economic policies, such as taxation, exchange rates, or visa regulations, can lead to cautious consumer behavior and reduced willingness to travel, particularly for discretionary spending on leisure tourism. As a result, individuals may opt to forgo travel or choose less expensive destinations and activities during uncertain economic times.

Several previous authors have investigated the relationship between tourism development and economic policy uncertainty but has not yielded consistent results. The opening for this research direction is that Baker, Bloom and Davis [9] developed Economic Policy Uncertainty (EPU) as a proxy for movements in policy-related economic uncertainty. Finding a one-way effect from EPU to TO is quite common in many contexts, while the opposite effect is quite scarce. Inefficiencies and sustainability in economic policies lead to risks to price stability and exchange rate, two factors in the tourism demand function. Tsui, Balli et al. [10] explored how the EPU affects business tourism in New Zealand. Uncertain economic policy in New Zealand reduces the number of business visitors from the country's main trading partners. Singh, Das et al. [28] found a relationship between EPU (domestic and global) and monthly international tourist arrivals to the United States. They found that the EPU had a slight immediate impact on tourist arrivals. Furthermore, the US EPU and the global EPU have a significant impact on the tourism industry, while the country-specific EPU's impact dominates the global EPU. This result is confirmed to be more common in developing economies than in developed countries. In India, economic policy uncertainty does not have much of an impact on tourism development in the country compared to geopolitical risks. With a larger number of observations of 92 countries, an increase in the global EPU leads to a decrease in the number of foreign tourists and an increase in spending.

Khan, Su et al. [11] investigated the causal relationship between uncertainty in economic policy and the tourism industry, showing that the EPU restricts domestic tourism while domestic tourism has a positive effect on the EPU in a given period. When comparing the domestic and foreign economic sectors in 124 countries, Nguyen, Thanh and Nguyen [13] concluded that increased economic instability increases domestic tourism and decreases tourism. Abroad, contrary to the views of some other scholars. From another perspective, by using the ARDL and non-linear ARDL approaches, the results of Sharma [29] confirmed that economic instability has an asymmetric impact on tourism demand. In particular, an increase in uncertainty has more negative effects than good effects caused by an improvement in the certainty of economic policies. In stark contrast to the above conclusions, Payne, Topcu et al. [14] found that there was no Granger causality from the EPU (global and US) to US outbound travel.

The comments made in the field of study are as follows. First, the figures used in the study should be for the period of months instead of years. Doubts were raised about the appropriateness of using annual data when Demir and Gözgör [30], Madanoglu and Ozdemir [31] attempted to examine the association between EPU and tourism. Annual data is said to not reflect well when the EPU was developed to capture short-term uncertainty. A long-term (annual) analysis using these data may not be entirely feasible because short-term variations are resolved over long periods. Second, because of the lack of diversity in understanding whether the impact on the direction of tourism development causes instability in economic policy, this study will address that problem by applying Singapore data from January 2003 to February 2022.

## Tourism development and government revenue

All around the world, the tourism and travel industries play a significant economic role. Proof that the tourism sector continues to be a substantial source of income and employment in

both official and informal areas in many nations [32]. Since tourism is a labor-intensive business, a great of local people are needed, and the majority of jobs don't require much formal training [33].

Besides, government revenues from the tourism sector can be categorized as direct and indirect contributions. Direct contributions are generated by taxes on incomes from tourism employment, and tourism businesses and by direct charges on tourists such as ecotax or departure taxes. Indirect contributions derive from taxes and duties on goods and services supplied to tourists. Governments at all levels (provincial/territorial and municipal) raise tax revenues from the various activities of tourists. For instance, when a tourist pays for a hotel room, this generates a goods and services tax, a provincial sales tax, and a room tax for the various levels of government. In addition, income taxes are collected from the earnings of hotel employees and from the profits of the business enterprise itself.

Governments collect revenues mainly for two purposes: to finance the goods and services they provide to people and businesses, and to fulfill their redistributive role. Comparing levels of government income across countries indicates the importance of the government sector in the economy in terms of the financial resources available so that countries can make use of all resources to develop the tourism sector.

## Tourism development and industrial production index

The industrial production index (IPI) represents the output and activity of the industry sector. The tourism industry develops in association with other related industries. Studies on this nexus are still limited and even contradictory. Hailemariam and Ivanovski [15] investigated the impact of world industrial production (WIP) on tourism net exports (TNXs) by applying an identified structural vector autoregression model over monthly data spanning from January 1999 to October 2020. Their findings revealed that a positive shock in WIP has a significant positive impact on demand for TNXs. In contrast, in the study of Uzuner and Ghosh [34], the asymmetric Granger causality between tourist arrivals and the world pandemic uncertainty index is tested by controlling for the industrial production index during the period from January 2000 to January 2020 in Italy. Interestingly, the negative shock in industrial production caused an increase in tourist arrivals.

In fact, during a vacation or visit to another nation, a tourist is seen as having an amazing range of needs for products and services. The demands of tourists are covered in a wide range of industries, including manufacturing, telecommunications, sewage, banking, medical and dental care, security, transportation, lodging, and one or two other necessities. Some business owners, such as a hotelier, perceive themselves as primarily engaged in the tourism industry from the supply side's point of view. Nevertheless, they are a part of the tourist business during the time that they are working to satisfy the wants of travelers. For example, economies driven by tourism make better use of their resources in the hospitality industry because of their focus on services [35]. The demand for hospitality services frequently fluctuates more sharply due to seasonal tourist flows. Specifically, the provision of hotels or different forms of accommodation to accommodate the growing number of tourists coming to the country.

Alternatively, in the field of healthcare, Lee [36] identified a uni-directional causal relationship in the long run from healthcare to international travel in Singapore. Policymakers consider healthcare facilities to be one of the important destination attributes for tourists, and the Singapore government has decided to establish it as a leading medical centre, thereby attracting more domestic and foreign tourists. Nowadays, health tourism has expanded to incorporate leisure activities and well-being improvement because patients now need time to rest and

recover. Destination marketers are required to reconsider why healthcare travelers choose to visit a particular location.

## 3. Research model and econometric strategy

Classical econometric methods, such as ordinary least square and ARDL approaches, only provide unknown estimated coefficients but are fixed. That means it ignored information contained in time-frequency domains, which raises doubt about the effectiveness of suggested policy implications [16, 37, 38]. This study applied two novel econometric approaches, including the Wavelet tool and the time-varying causality method, to overcome this limitation. The Wavelet methodology was proposed in the mid-1980s, allowing the decomposition of the time series into several wavelet frequencies. Thus, it extracts localized information in both the time and frequency domains, and therefore it captures the complete information contained in time series specific to location-scale domain. In the first step, continuous wavelet transform (CWT) is employed to decompose the concerned series into wavelets. The CWT of a given time series $x(t)$ can be specified as:

$$W_x(s) = \int_{-\infty}^{\infty} x(t) \frac{1}{\sqrt{s}} \psi^* \left(\frac{t}{s}\right) dt \qquad (1)$$

where $^*$ denotes the complex conjugate, while $\psi^* \left(\frac{t}{s}\right)$ reveals the complex conjugate function of $\psi \left(\frac{t}{s}\right)$, namely the so-called basis wavelet function. And, $s$ is the wavelet scale that controls how the mother wavelet is stretched. Based on the continuous wavelet transform, Torrence and Webster [39] suggested that the cross-wavelet transform technique of two series $x(t)$ and $y(t)$ can be given as:

$$W_n^{XY}(u, s) = W_n^X(u, s) W_n^{Y*}(u, s) \qquad (2)$$

where $u$ presents the position and $s$ denotes the scale. Consequently, the equation of the squared wavelet coherence can be specified as:

$$R_n^2(u, s) = \frac{\left| S\left(s^{-1} W_n^{XY}(u, s)\right)\right|^2}{S\left(s^{-1} |W_X(u, s)|^2\right) S\left(s^{-1} |W_Y(u, s)|^2\right)} \qquad (3)$$

where, $S$ connotes the smoothing process for time and frequency at the same time. The study follows the suggested scales by Torrence and Webster [39], in which 2 to 8-month scale presents the short-run horizon. Similarly, 8 to 32 months refers to medium-term horizon, and the period above 32 months demonstrates the long-run horizon. To capture the causal association between positive or negative dependency, Torrence and Webster [39] suggested the equation of the phase difference mechanism between $x(t)$ and $y(t)$ series as (labeled XWT):

$$\phi_{XY}(u, s) = \tan^{-1} \left( \frac{\Im\{S(s^{-1} W_{XY}(u, s))\}}{\Re\{S(s^{-1} W_{XY}(u, s))\}} \right) \qquad (4)$$

where, $\Im$ and $\Re$ are the imaginary and real parts of the smooth power spectrum, respectively. The direction of causality and correlation is indicated by the arrows. Two-time series will move simultaneously at a given frequency if their phase difference is zero. If $\phi(u, s) \in [-\pi/2; 0]$ and $\phi_{XY}(u, s) \in [0; \pi/2]$, then the two series are in phase with $x(t)$ is leading $y(t)$, and $y(t)$ is leading $x(t)$, respectively. On the other hand, an equivalent to negative covariance is defined as an anti-phase relation when $\phi_{XY}(u, s) \in [\pi\pi/2; \pi] \bigcup [\pi\pi/2; \pi]$. Finally, if $\phi_{XY}(u, s) \in [-\pi; -\pi/2]$ the $y(t)$ is leading $x(t)$, while $x(t)$ is leading $y(t)$ when $\phi_{XY}(u, s) \in [\pi/2; \pi]$.

To robust the Wavelet findings, the time-varying causality approach introduced by Shi, Hurn and Phillips [40] will be applied. This method used several VAR-based methods to gage time-varying causality, including the "forward expanding window", "rolling window", and "recursive evolving window" approach. In this study, we only use the "rolling window" approach to robust the Wavelet findings. The equation of "rolling window" approach is defined as:

$$\hat{f}_e = \frac{\inf}{f \in [f_0, 1]} \left\{ f : w_f(f - f_0) > cv \right\} \quad \text{and} \quad \hat{f}_f = \frac{\inf}{f \in [\hat{f}_e, 1]} \left\{ f : w_f(f - f_0) < cv \right\} \quad (5)$$

wheres, $cv$ denotes the $Wf$ statistics's critical value, and $\hat{f}_e$ and $\hat{f}_f$ present the first (chronologically estimated) observation, the estimated initial temporal events when the t-statistics pass or drop under the significant values for the causal relationship. The null hypothesis is postulated that: TO does not Granger-cause EPU, GR, and IPI, respectively. The null hypothesis is rejected if the test static sequence is greater than 5% critical value [41]. All data of variables are collected from the National Statistics Office of Singapore. The lengthened data is January 2003 —February 2022.

## 4. Empirical results

### Descriptive statistics

The empirical results have now been presented. Fig 1 illustrates a significant volatility in the tourism industry, specifically in 2011 and from 2020 to 2022. This phenomenon has resulted in a decrease in government revenue and an increase in economic policy uncertainty. To be more specific, the total number of international arrivals in December 2019 reached 1.7 million passengers, but plummeted to a mere 95 passengers by September 2020. This decline was due to the Singapore government's decision to implement a nationwide lockdown in response to the COVID-19 pandemic. Concurrently, government revenue also experienced a decline from 7.051 billion Singapore dollars to 5.533 billion dollars. Throughout the sample period, these four variables consistently exhibited a similar level of volatility.

### Unit root test

According to Phiri, Anyikwa and Moyo [18], unit root test results are important as the time-varying causality tests require the series to be stationary. Hence, the two famous tests proposed by Dickey and Fuller [42], and Phillips and Perron [43], were employed. Table 1 revealed that four variables are stationarity at the level difference I(0), implying that the condition of the time-varying causality technique is satisfied.

### The cross-wavelet (CWT) and wavelet coherence (XWT) findings

In the next step, the study applied the CWT and XWT analysis to illustrate the local covariance and lead-lag structure between TO and three variables (EPU, GR, IPI) across different scales and periods. The color scheme is as follows: yellow indicates high power, while blue represents low power. Additionally, warmer shades of yellow indicate high joint power between the two variables, whereas cooler shades of blue suggest low power between them [44, 45]. Accordingly, Fig 2 reveals a robust causal correlation between EPU and TO at medium-high frequencies. This implies that EPU exhibits similar volatility to TO in the medium- and long-term. Furthermore, Fig 3 demonstrates the co-movement and lead-lag structure between EPU and TO in Singapore, as revealed by the XWT analysis. Notably, the arrows point left and upward at scales of 0–8 months, indicating EPU as the leading variable. A rise in EPU leads to a decline

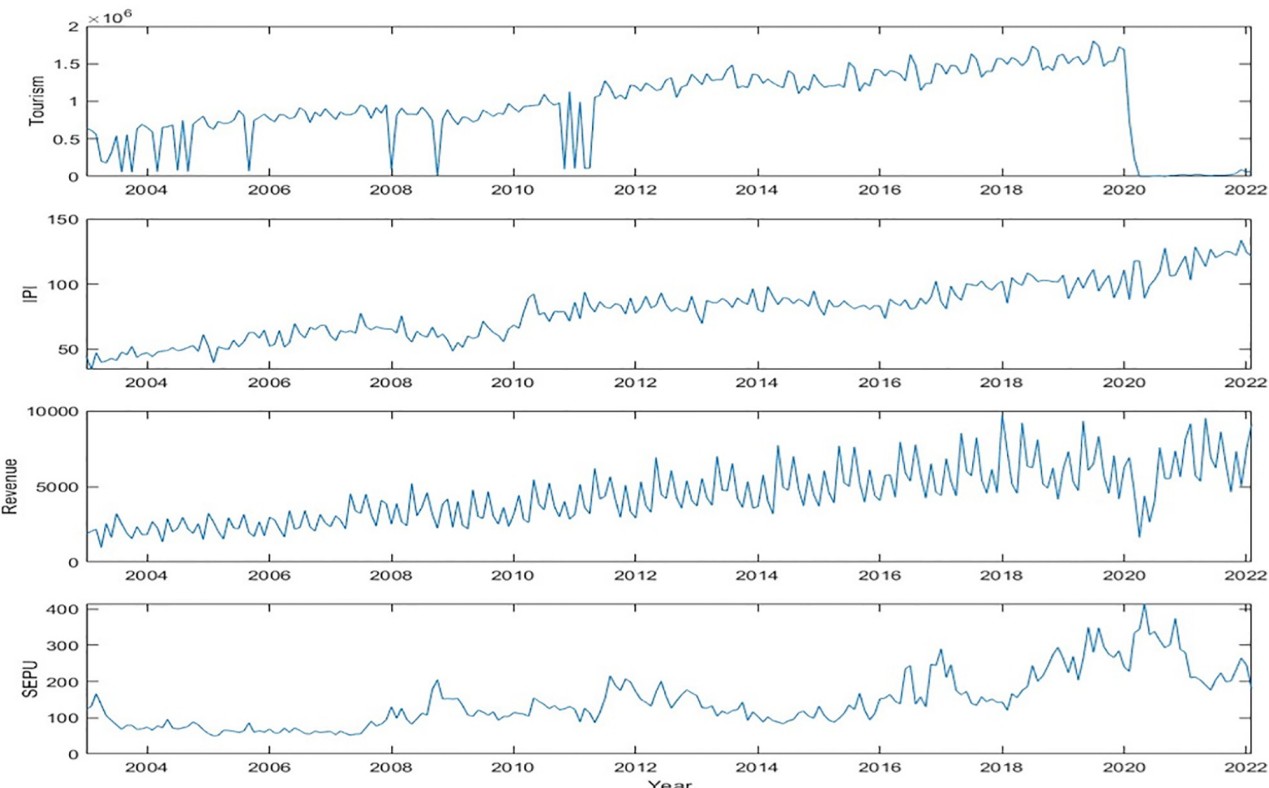

**Fig 1. The volatility of variables from Jan 2003 to Feb 2022.**

in TO at low frequencies, suggesting that EPU negatively influences TO in the short term. Additionally, Fig 3 highlights that the arrows point right and upward at medium frequencies (16 to 64-month scales) and within the time frame of February 2013 to May 2018. This indicates that TO acts as the leading variable, where an increase in TO decreases EPU during 2013–2015. Overall, the relationship between EPU and TO is significant in the short- and medium-term.

Similarly, Figs 4 and 5 denote the causal association between TO and GR variables. Most arrows are right, implying that two variables are in phase. Accordingly, the correlation between TO and GR is weak in the short term when the island is blue. Fig 5 reveals that TO was only strongly related to GR from 2003 to 2013 at high frequencies. More precisely, an

**Table 1. The unit root test result.**

| Variables | Augmented Dickey-Fuller test | | Phillips-Perron test | |
|---|---|---|---|---|
| | I(0) | I(1) | I(0) | I(1) |
| TO | -2.145 | -24.897*** | -3.224* | -31.568*** |
| EPU | -4.228*** | -19.036*** | -5.099*** | -23.138*** |
| GR | -3.607** | -6.729*** | -16.130*** | -99.042*** |
| IPI | -2.352 | -11.619*** | -11.708*** | -102.495*** |

Note: Two tests are chosen based on the Akaike Information Criterion with intercept and trend.

***, ** and * respectively denote significance levels of 1%; 5% and 10%.

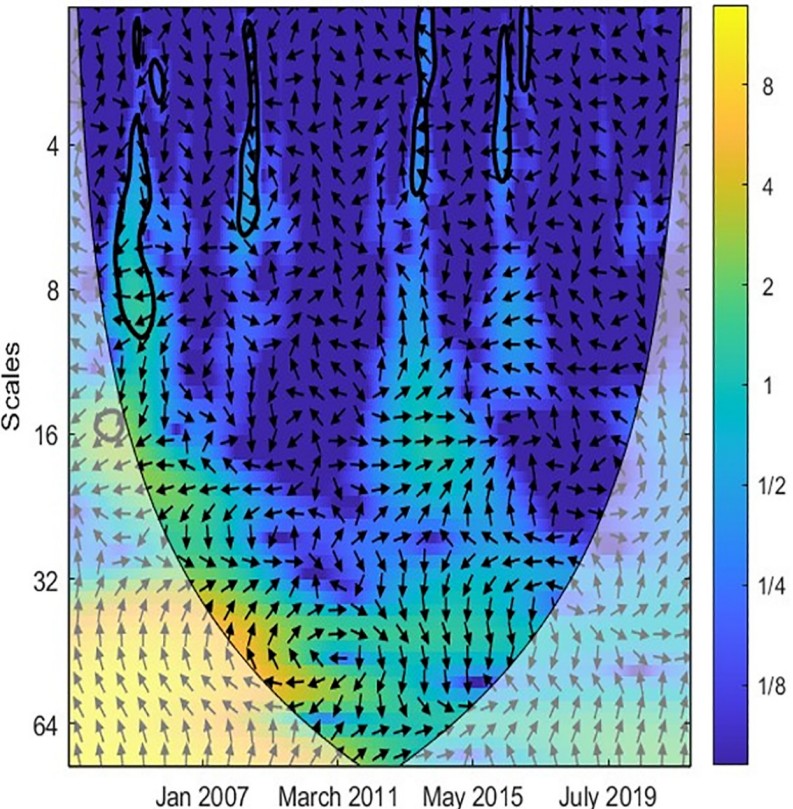

**Fig 2. CWT between TO and EPU.** Note: The color bar shown on the right side of each figure suggests the color code for power that ranges from blue (low) to yellow (high). The X-axis is the study time period and Y-axis indicates frequencies.

increase in TO leads to an increase in GR. From 2010 to now, the correlation between TO and GR is unclear. This implies that Singapore's government has successfully diversified revenue sources and reduced reliance on the tourism industry. Likewise, Figs 6 and 7 document the interaction between TO and IPI. In more detail, Fig 7 shows that a negative causal correlation between TO and IPI is only firm in the long term. The arrows are right and upward, implying TO is the leading variable, and an increase in TO might lead to a decrease in other industries. However, this conclusion is only valid from March 2011 to January 2016. In general, the TO correlates weaker with IPI than EPU and GR variables. A summarized result of Wavelet coherence analysis is presented in Table 2.

## The time-varying causality results

As mentioned in the methodology section, this study employs the time-varying causality method to enhance the robustness of the wavelet findings. The article illustrates the results obtained from the "rolling window" approach. Notably, Fig 8 demonstrates a strong uni-directional causality from TO to EPU in three periods: January 2003 to February 2011, January 2016 to December 2016, and December 2019 to February 2022. Similarly, Fig 9 reveals a strong causal association between TO and GR from January 2016 to February 2022. Additionally, there is no causality between TO and IPI between January 2003 and July 2015 (see Fig 10). However, a significant causal relationship between these variables becomes evident from May

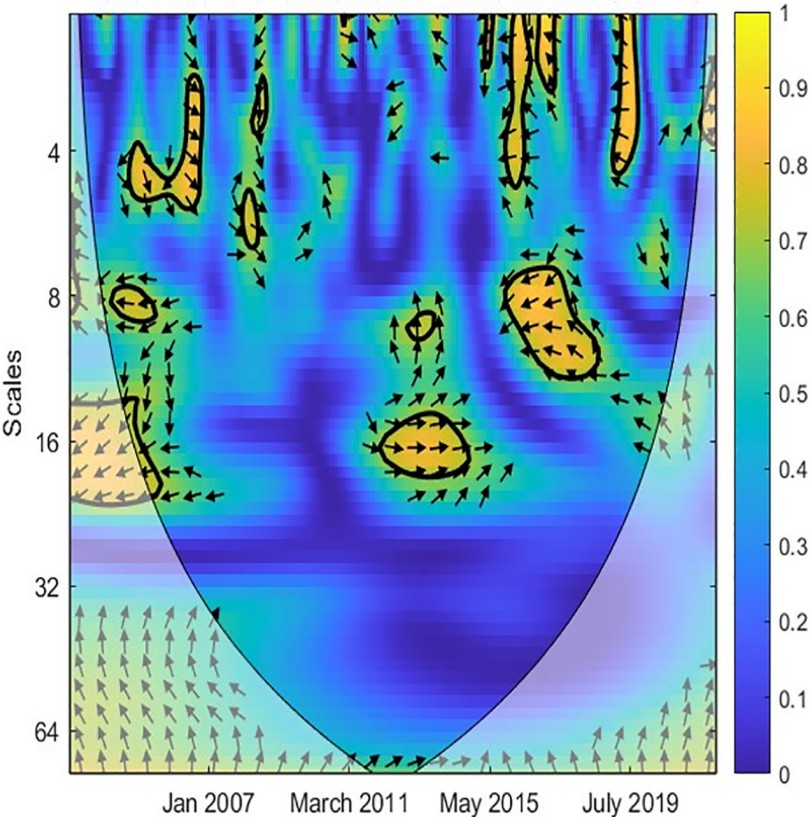

**Fig 3. XWT between TO and EPU.** Note: The time-scale wavelet coherence levels are captured are represented by a colour spectrum, with yellow indicates high power, while blue represents low power. The arrows themselves, represent the phase difference dynamics between the time series. The right (left) pointed arrow denotes phase-in (phase-out) which can be interpreted as a positive (negative) correlation between the series.

2015 to October 2017, and from September 2019 to February 2022. Overall, these three figures highlight the surge in EPU during the COVID-19 era and its impact on tourism development in Singapore.

Despite certain differences, the results obtained from the time-varying causality technique align with the findings from the cross-wavelet coherence analysis, particularly regarding the relationship between TO and EPU. Overall, the TO variable demonstrates a strong correlation with EPU, a moderate correlation with GR, and a weaker correlation with the IPI variable.

## 5. Discussion

The results of this study shed light on the nexus between economic policy uncertainty, government revenue, industrial production index, and tourism development in Singapore from January 2003 to February 2022. Despite the abundance of previous studies, the relationship between EPU and tourism development still has much attention from economists and policy-makers worldwide. By applying Wavelet tools, the findings of this study confirmed a robust negative relationship between EPU and TO in Singapore from January 2003 to February 2022 at different time-frequency scales. More precisely, economic uncertainty negatively affects tourism development at low frequencies from February 2013 to August 2018. In contrast, in the long term, the Wavelet coherence plots revealed that an increased tourism development

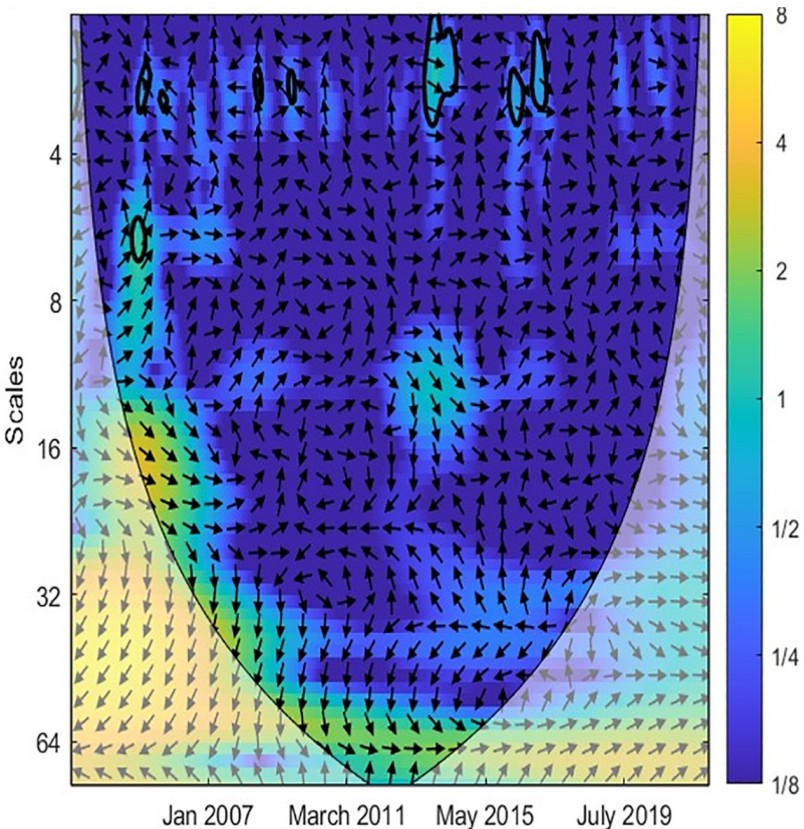

**Fig 4. CWT between TO and GR.**

leads to decreased EPU. The findings align with the conclusions of Wu and Wu [46] for BRIC countries and Singh, Das et al. [28] for the United States economy. Singapore's economy is highly interconnected with the global economy. Events such as the Global Financial Crisis (2008–2009), the European Debt Crisis (2010–2012), and the COVID-19 pandemic (2020 onwards) likely contributed to periods of heightened economic policy uncertainty in Singapore. Uncertainties surrounding trade tensions between major economies, geopolitical events, and shifts in global economic trends could also influence economic policy uncertainty in Singapore. When facing economic uncertainty, the priority choice of individuals and enterprises is decreasing investments and reducing outbound tourism expenditures, which negatively affect tourism investments, hindering the growth of the tourism sector. An intriguing finding the authors realize is that increasing tourism development reduces EPU.

Some reasons are suggested to explain the findings: (i) In the short term, tourism is susceptible to various forms of uncertainty, such as conflicts, terrorism, natural catastrophes, epidemics, contagions, and financial crises, which can negatively impact tourism demand [47, 48]. (ii) In the long term, developing the tourism industry will stimulate the diversification of national revenue sources. When the economy depends on many sectors, fluctuations in a specific sector will not strongly impact the entire economy [33]. Additionally, developing the tourism industry also strengthens local economies [49], boosts exporting [50], enhances national image [37], and creates community stability [51]. Hence, many economists and policymakers believe that boosting tourism development is crucial to combat economic uncertainty.

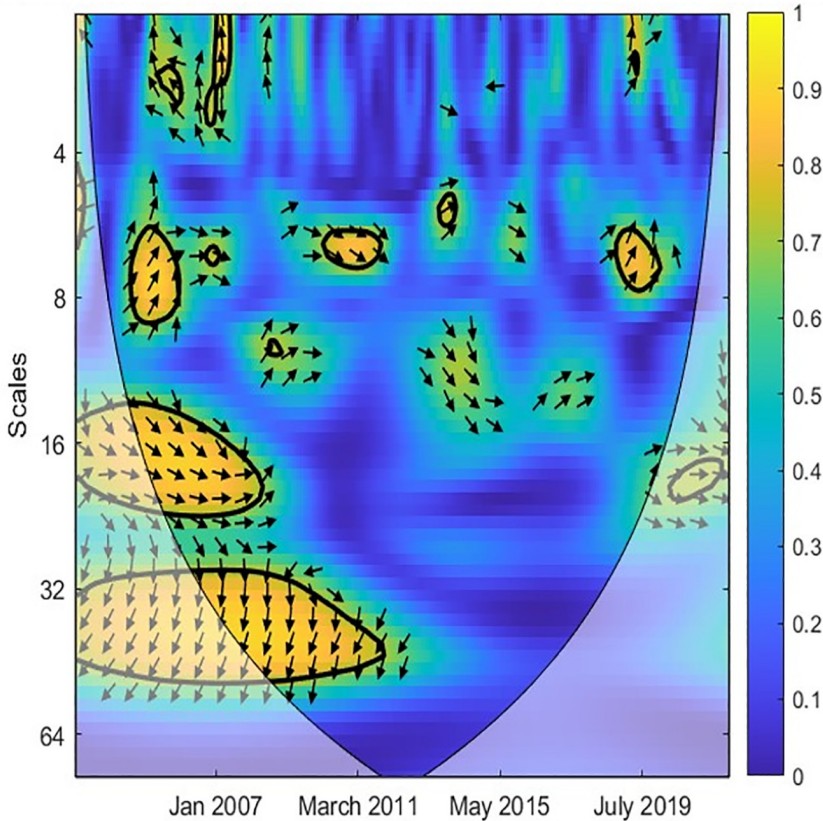

**Fig 5. XWT between TO and GR.**

Another important finding of this study is that tourism development can lead to a reduction in the development of other sectors in Singapore in the short- and medium-term. In contrast, TO tends to enhance IPI in the long term. Significant investments in tourism infrastructure, such as hotels, entertainment facilities, and tourist attractions, may divert resources away from other sectors [52]. This allocation of resources could limit funding available for the development of other industries. Besides, a heavy reliance on tourism revenue may lead to a disproportionate focus on catering to tourists' needs and preferences at the expense of other sectors. This focus can skew economic priorities and neglect the development of diverse industries. Fortunately, this conclusion is only valid for the period 2003–2015. According to the World Travel & Tourism Council [53], the Singaporean government prioritized investments and resources in the tourism sector at the expense of other sectors from 1978 to 1992. This results in insufficient funding for areas such as education, healthcare, or infrastructure. Realizing this limitation, after 2015 the Singapore government applied many measures to diversify the economy, focusing on digital transformation, high technology, and financial services. These policies help Singapore better respond to risks and promote the supporting role of tourism development for other sectors. This conclusion is supported by Li, Duan et al. [54], who confirmed that tourism development efficiency of 58 China's cities is increasing year by year.

In addition, the authors discover that initially, a rise in tourism activity corresponds to a rise in government revenue. However, the correlation between tourism and government revenue has become ambiguous after January 2013. This suggests that the Singaporean government has effectively broadened its revenue streams and diminished its dependence on the tourism

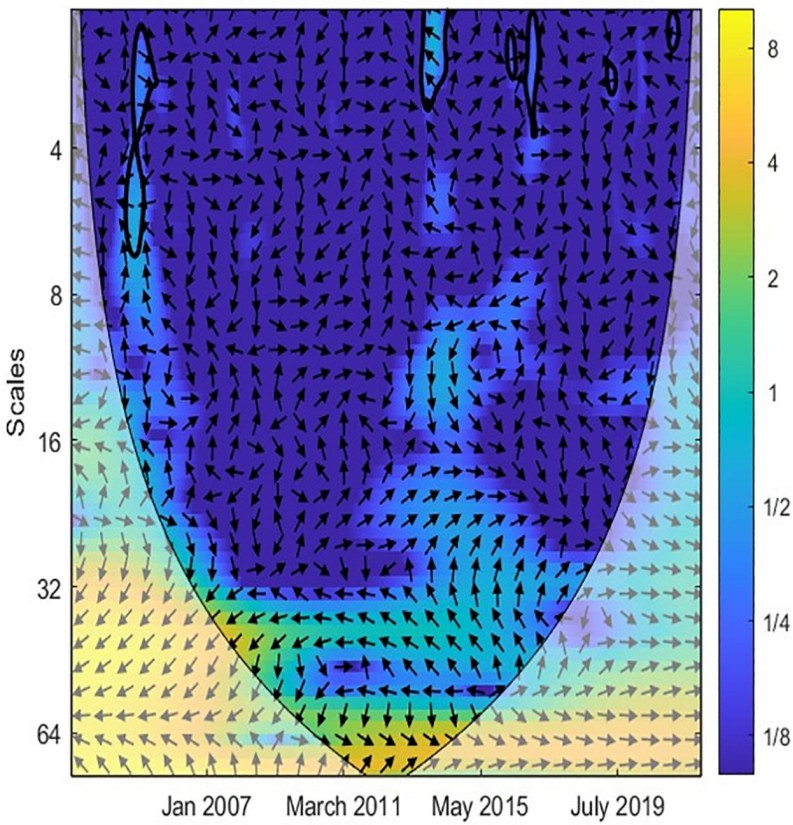

**Fig 6. CWT between TO and IPI.**

sector. In other words, while tourism continues to contribute to government revenue, other sectors or sources of revenue have become increasingly important. This diversification strategy is beneficial because it reduces Singapore's vulnerability to fluctuations or downturns in the tourism industry. It shows that Singapore's economy has matured and diversified, allowing the government to maintain stable revenue streams even when tourism may not be performing as strongly. By investing in and promoting other sectors, such as finance, technology, and manufacturing, Singapore could have expanded its revenue sources, leading to a weaker correlation between tourism and government revenue. This outcome contrasts sharply with viewpoint of Gnangnon [55]. The empirical findings demonstrate that international tourism receipts have a notable and beneficial influence on non-resource tax revenue from 1995 to. Moreover, this influence strengthens as countries progress in development, suggesting that advanced economies benefit more from increased international tourism receipts in terms of non-resource tax revenue compared to less developed economies.

## 6. Conclusion and policy implications

The previous studies on the relationship between tourism development and economic policy uncertainty have largely ignored the distinction between short, medium, and long-term by decomposing examined variables at different time scales. By applying the Wavelet tools and time-varying causality method, this study confirms a strong relationship between economic policy uncertainty, government revenue, industrial production index, and tourism

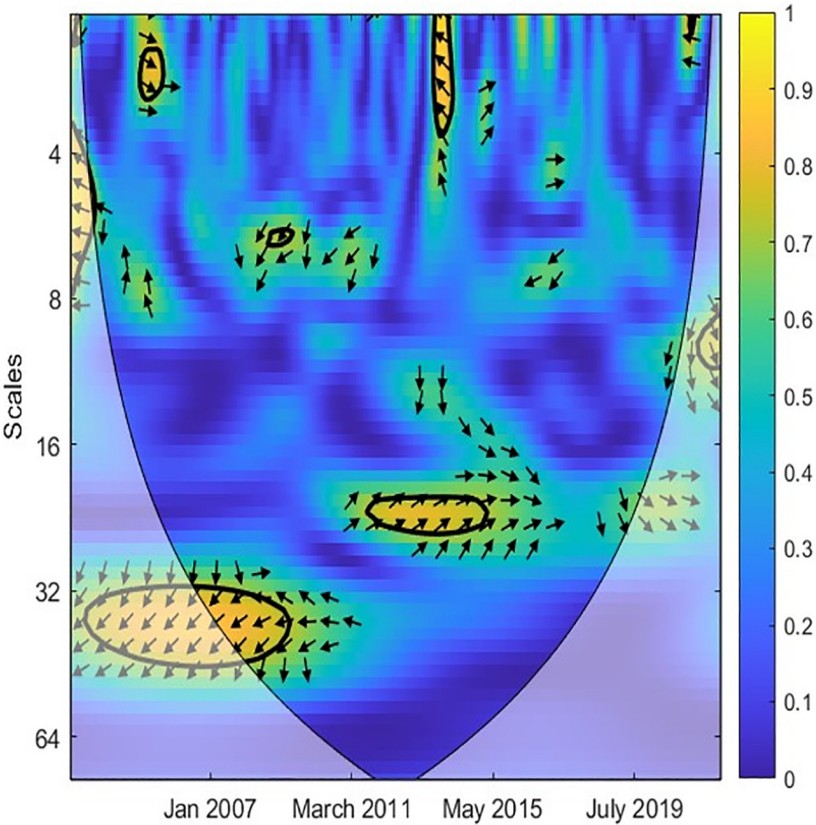

**Fig 7. XWT between TO and IPI.**

development in Singapore from January 2003 to February 2022. The results from wavelet analysis indicate that strong lead-lag structural interrelationships change over time, displaying low to medium-frequencies cycles between EPU and TO. Besides, the study also shows strong evidence of the causal relationships between tourism development, government revenue, and

**Table 2. Wavelet coherence findings summary.**

| Frequencies | Cross-wavelet coherence findings |
|---|---|
| TO-EPU | |
| High frequency | ↑TO → ↓EPU |
| Medium frequency | ↑TO → ↓EPU, ↑EPU → ↑TO |
| Low frequency | ↑EPU → ↓TO |
| TO-GR | |
| High frequency | ↑TO → ↑GR |
| Medium frequency | ↑TO → ↑GR, ↑TO → ↓GR |
| Low frequency | ↑GR → ↑TO |
| TO-IPI | |
| High frequency | ↑TO → ↑IPI |
| Medium frequency | ↑TO → ↓IPI |
| Low frequency | No causality |

Note: ↑ reflects an increase in, ↓ presents a decrease in, → denotes the variable on the left side of the arrow leads the variable on the right side's arrow.

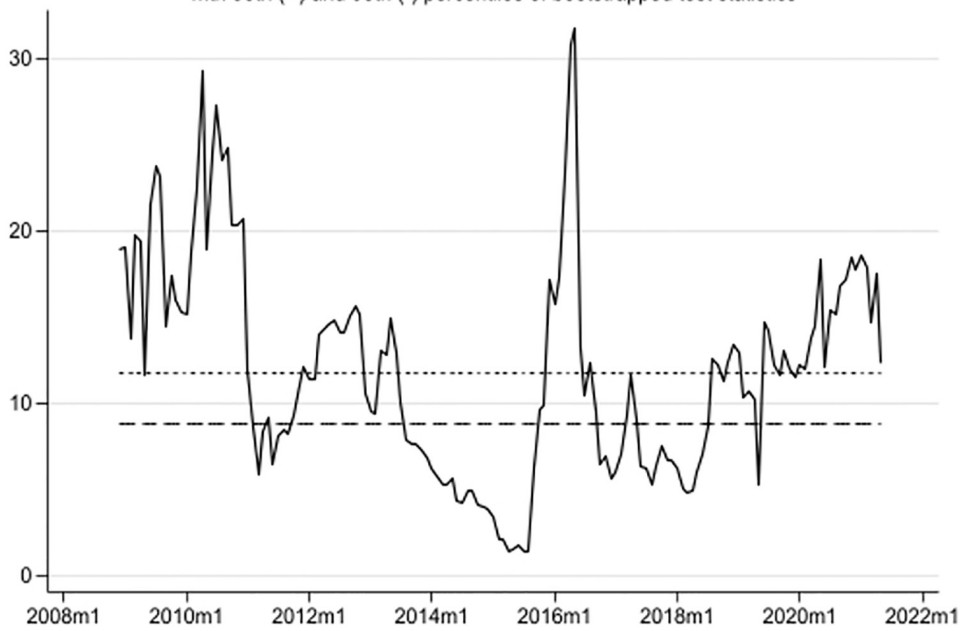

**Fig 8. TO to EPU time-varying causality.**

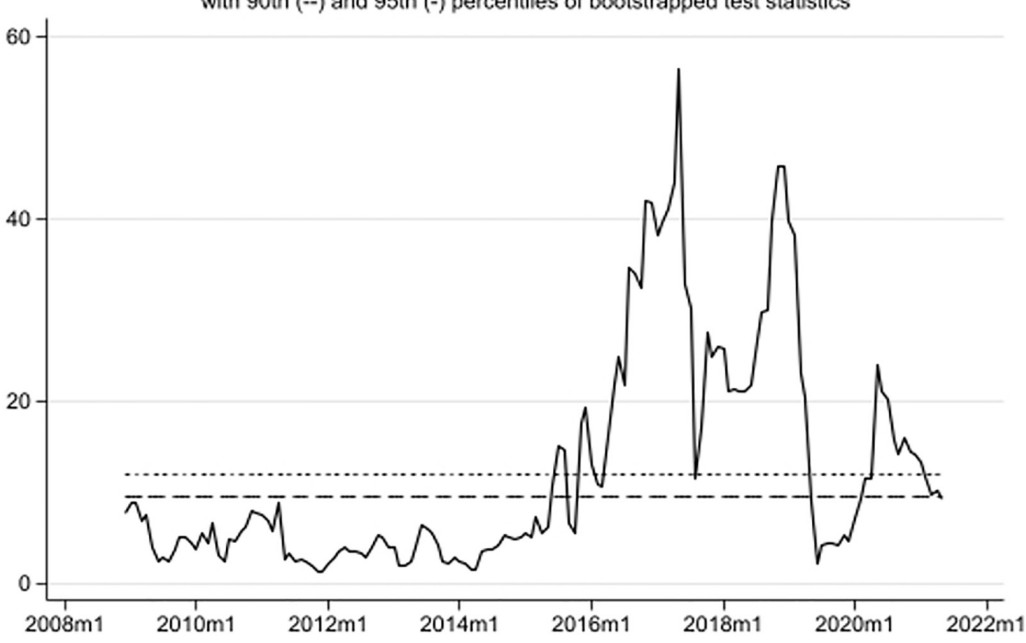

**Fig 9. TO to GR time-varying causality.**

**Fig 10. TO to IPI time-varying causality.**

industrial production index in Singapore. More precisely, the tourism industry's development only negatively impacted other sectors before January 2016. After this period, tourism development will support the development of other industries, thereby reducing economic instability.

Based on the findings, some policy implications are suggested:

- First, uncertainty can lead to a decrease in tourist arrivals due to concerns about economic stability. Governments may need to allocate more resources to tourism promotion and marketing campaigns to mitigate the negative impact of economic uncertainty on tourist demand. Clear and consistent messaging about the attractiveness and safety of the destination can help reassure potential visitors. Policymakers should develop risk management strategies to cushion the tourism sector from the effects of economic policy uncertainty. This may include creating contingency plans, diversifying tourism products and markets, and providing financial support to tourism businesses during periods of economic instability. Economic policy uncertainty often arises from various sources, such as political instability, trade tensions, and regulatory changes. Therefore, collaboration between the public and private sectors is crucial for promoting tourism development in the face of economic policy uncertainty. Governments can work with industry stakeholders to identify challenges, formulate solutions, and implement policies that support tourism growth. Public-private partnerships can also facilitate the sharing of resources and expertise to address common concerns.

- Second, policymakers should focus on diversifying revenue sources beyond tourism to mitigate the impact of fluctuations in the tourism sector on economic policy uncertainty. This

can involve promoting investment in other sectors such as technology, manufacturing, and finance to reduce reliance on tourism-related revenue. Moreover, enhancing institutional frameworks and regulatory mechanisms can help manage economic policy uncertainty associated with tourism development. This includes establishing transparent and stable policies, regulations, and enforcement mechanisms to provide clarity and predictability for businesses and investors in the tourism industry. Adopting a long-term perspective and prioritizing sustainability in tourism development can help reduce economic policy uncertainty and promote the industry's resilience to external shocks. Governments should develop comprehensive tourism strategies that balance economic growth with environmental conservation, cultural preservation, and social equity to ensure the industry's long-term viability.

- Third, governments should explore ways to promote synergies between tourism development and industrial production to maximize economic benefits. This may include supporting the development of tourism-related manufacturing activities, such as the production of souvenirs, crafts, and equipment, to create additional opportunities for industrial growth and job creation. Meanwhile, investing in infrastructure that supports both tourism development and industrial production can help strengthen the linkages between the two sectors. Governments should prioritize investments in transportation, logistics, and utilities to improve connectivity and accessibility for tourists and industrial producers alike. In addition, enhancing skills development and training programs in areas relevant to both tourism and industrial production can help ensure a qualified workforce to support growth in both sectors. Governments should collaborate with industry stakeholders to identify skill gaps and develop training initiatives that address the needs of both sectors.

- Finally, diversification of revenue streams beyond tourism is essential to reduce vulnerability to fluctuations in this sector. Governments should periodically review and adjust fiscal policies to ensure that tourism-related revenue is effectively captured and utilized for public services and infrastructure development. This may include implementing targeted taxes or levies on tourism activities to generate additional government revenue while minimizing negative impacts on tourism demand. Furthermore, encouraging sustainable tourism practices can help maximize government revenue from tourism while minimizing negative social and environmental impacts. Policymakers should implement regulations and incentives to promote responsible tourism behavior, conservation of natural resources, and cultural preservation.

Several important findings on the nexus between economic policy uncertainty, government revenue, industrial production index, and tourism development in Singapore have been confirmed by this study. However, this work also has its constraints. The current research is focused on Singapore, a top-visited country, and there is potential for extending its scope to explore analogous issues in various developing economies, aiming to improve our comprehension of the subject. Furthermore, to paint a comprehensive picture of the connection between economic uncertainty and tourism development, it is imperative to delve deeper into research on contributors to uncertainty and instability, such as institutional quality and financial development. This would provide a more holistic understanding of the interplay between economic uncertainty and economic activities./.

## Acknowledgments

We thank the Editor-in-Chief, Senior Editor, and anonymous reviewers for their useful comments and suggestions. All remaining errors are ours.

## Author Contributions

**Conceptualization:** Canh Chi Hoang, Nguyen Huynh Mai Tram.

**Data curation:** Canh Chi Hoang.

**Formal analysis:** Nguyen Huynh Mai Tram.

**Methodology:** Bui Hoang Ngoc.

**Project administration:** Nguyen Huynh Mai Tram.

**Resources:** Canh Chi Hoang.

**Software:** Bui Hoang Ngoc.

**Supervision:** Bui Hoang Ngoc.

**Writing – original draft:** Canh Chi Hoang, Nguyen Huynh Mai Tram.

**Writing – review & editing:** Bui Hoang Ngoc.

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
