## [Decision Letter · Decision Letter 0]

7 Feb 2024

PONE-D-24-00197A TIME-VARYING ANALYSIS BETWEEN ECONOMIC UNCERTAINTY AND TOURISM DEVELOPMENT IN SINGAPORE.PLOS ONE

Dear Dr. Ngoc Bui Hoang,

Thank you for submitting your manuscript to PLOS ONE. After careful consideration, we feel that it has merit but does not fully meet PLOS ONE’s publication criteria as it currently stands. Therefore, we invite you to submit a revised version of the manuscript that addresses the points raised during the review process.

We look forward to receiving your revised manuscript.

Kind regards,

Ricky Chee Jiun Chia

Academic Editor

PLOS ONE

Journal Requirements:

Reviewers' comments:

Reviewer's Responses to Questions

**Comments to the Author**

1. Is the manuscript technically sound, and do the data support the conclusions?

Reviewer #1: Partly

Reviewer #2: Partly

2. Has the statistical analysis been performed appropriately and rigorously? 

Reviewer #1: Yes

Reviewer #2: I Don't Know

3. Have the authors made all data underlying the findings in their manuscript fully available?

Reviewer #1: No

Reviewer #2: Yes

4. Is the manuscript presented in an intelligible fashion and written in standard English?

Reviewer #1: Yes

Reviewer #2: No

5. Review Comments to the Author

Reviewer #1: The paper examines the relationship between economic uncertainty, government revenue and industrial production on tourism development in Singapore using wavelet coherence analysis and time-varying causality techniques. Whilst the study is OK, there are some improvements that need to be made to make of publishable quality.

The abstract is too plain and needs to inform the reader more on the obtained results and policy implications/importance of the study.

The introduction

In the literature review, whilst the review of empirical studies is sufficient, the theoretical foundations of the study is missing.

The methods need to be described in more detailed. For instance, in the wavelet coherence analysis, what is the ‘mother wavelet’ that is being used and why was this mother wavelet chosen? Also, the time-varying causality tests need to be described in greater detail. What are the null hypothesis of the causality tests and what criteria are used for rejecting or accepting the tested hypotheses?

In the empirical results, the authors need to show the results from the wavelet power spectrum plots to give an indication the time-frequency evolution of the individual time series which would be of interest to the readers. This can be provided together with some basic time series statistics for the variables. Also, unit root test results are important as the time-varying causality tests require the series to be stationary. Collectively, the authors should include a new section addressing these issues.

The wavelet coherence plots do not report the actual dates on the horizontal axis. The authors need to do so as it is difficult to interpret the wavelet plots without knowing when ‘certain events’ took place.

The comparison between the wavelet coherence plots and time-varying causality tests has not been done properly. What are the common findings and what findings differ? Are there an explanations for the differences in the findings.

In the further discussion of the results, much emphasis was place on the economic uncertainty-tourism relationship and not much was said about the government- expenditure-tourism or industrial production-tourism relationships. The authors need to discuss these three speres collectively. Same applies to the conclusion of the paper.

The paper could also do with some proof reading to eliminate editorial mishaps.

Reviewer #2: Main comments:

1) In the introduction it is not clear if authors are considering impact of tourism on policy uncertainty or vice versa. In general, from the introduction, the same lack of clarity involves all the causality links they are investigating. It should be clarified from the start that they are exploring both directions of causality for all three of the investigated variables as this appears to be their main contribution.

2) In the introduction the order of issues as presented in points i), ii) and iii) in the introduction is not consistent with the more detailed description that follows.

3) Since the dataset also includes the Covid-19 pandemic period, the authors should better comment on how this may affect their results. For example, could the fact there are no long-term relationships between TO and EPU be a consequence of the inclusion of pandemic data? Perhaps running the analysis excluding 2020-22 could be a robustness check.

4) Sometimes the authors use economic stability and EPU as synonyms, but they are not (example page. 7, last paragraph, line 8)

Comments on answers to drop-down menu questions:

1) I have answered “No” to the question “Is the manuscript technically sound, and do the data support the conclusions?” because of the Main Comment I raised regarding the inclusion of pandemic data in the sample.

2) I have answered “I don’t know” to the question: “Has the statistical analysis been performed appropriately and rigorously?” because I am not an expert in time-series analysis and in the wavelet methodology employed.

3) I have answered “No” to the question regarding whether the “manuscript presented in an intelligible fashion and written in standard English?” because the English needs polishing, and I suggest that the paper be proofread by a native English speaker before being published.

For Example:

Page 10, section 2, last paragraph, last line: “Destination marketers are required by evolution to reconsider why healthcare travelers choose to visit a particular location.” What does “by evolution mean”?

Also, there several typos:

For Example: page 15, par 1, line 6: “September 2029” should it be “2020”?

6. PLOS authors have the option to publish the peer review history of their article (what does this mean?). If published, this will include your full peer review and any attached files.

Reviewer #1: **Yes: **Andrew Phiri

Reviewer #2: No

---

## [Author Response · Author response to Decision Letter 0]

25 Mar 2024

The authors thank the two esteemed reviewers for their valuable comments. We regret some weaknesses of the original manuscript. In this revision, The authors have tried to address every comment and suggestion. All modified parts are indicated in green in the new version of the manuscript. The authors sincerely hope that the revised manuscript meets the expectations of the Reviewers.

---

## [Editor Report · Decision Letter 1]

16 Apr 2024

A TIME-VARYING ANALYSIS BETWEEN ECONOMIC UNCERTAINTY AND TOURISM DEVELOPMENT IN SINGAPORE.

PONE-D-24-00197R1

Dear Dr. Ngoc Bui Hoang,

We’re pleased to inform you that your manuscript has been judged scientifically suitable for publication and will be formally accepted for publication once it meets all outstanding technical requirements.

Kind regards,

Ricky Chee Jiun Chia

Academic Editor

PLOS ONE
---

## [Editor Report · Acceptance letter]

14 May 2024

PONE-D-24-00197R1 

PLOS ONE

Dear Dr. Ngoc, 

I'm pleased to inform you that your manuscript has been deemed suitable for publication in PLOS ONE. Congratulations! Your manuscript is now being handed over to our production team.

Kind regards, 

on behalf of

Dr. Ricky Chee Jiun Chia 

Academic Editor

PLOS ONE